# An Overview of Antimicrobial Resistance in Saudi Arabia (2013–2023) and the Need for National Surveillance

**DOI:** 10.3390/microorganisms11082086

**Published:** 2023-08-15

**Authors:** Abrar K. Thabit, Afaq Y. Alabbasi, Faris S. Alnezary, Imtinan A. Almasoudi

**Affiliations:** 1Department of Pharmacy Practice, Faculty of Pharmacy, King Abdulaziz University, 7027 Abdullah Al-Sulaiman Rd, Jeddah 22254-2265, Saudi Arabia; 2Eastern Health Cluster, Ministry of Health, Dammam 32253, Saudi Arabia; 3Department of Clinical and Hospital Pharmacy, College of Pharmacy, Taibah University, Madinah 41477, Saudi Arabia; 4Clinical Pharmacy Department, King Saud Medical City, Ministry of Health, Riyadh 12746, Saudi Arabia

**Keywords:** antimicrobial resistance (AMR), surveillance, gram-positive, gram-negative, broth microdilution (BMD)

## Abstract

Antimicrobial resistance (AMR) is a well-recognized global threat. The World Health Organization (WHO) issued a report ranking the critical types of bacterial resistance that need to be monitored. Several studies from individual institutions in Saudi Arabia have reported rates of antimicrobial resistance using automated methods. However, no national surveillance study has been conducted to date using gold standard methods for antimicrobial susceptibility testing. This review summarizes AMR rates for major bacterial pathogens in Saudi Arabia and provides a justification for the need for a national surveillance project. In Saudi Arabia, AMR rates for both Gram-positive and Gram-negative bacteria are on the rise. Surveillance studies help identify AMR trends and emergence of outbreaks. The WHO has started a program, the Global Antimicrobial Resistance Surveillance System (GLASS), encouraging its member states, including Saudi Arabia, to conduct antimicrobial surveillance studies to estimate AMR rates worldwide. Of the microbiological methods used to test antimicrobial susceptibility, only broth microdilution (BMD) is considered the “gold standard.” As AMR studies in Saudi Arabia are sparse, mostly limited to single centers and were conducted using automated methods, a national AMR surveillance project is needed to evaluate the current status and to inform stewardship decisions.

## 1. Introduction

Bacterial resistance to antimicrobial agents is an emerging global problem that is posing a threat and has a negative impact on clinical outcomes of patients infected with such organisms, as well as on the economic outcomes of hospitals and governments [1]. In 2017, The World Health Organization (WHO) released a list of antibiotic-resistant priority pathogens. The list is categorized as critical, high, and medium priority pathogens [2]. These multidrug-resistant (MDR) bacteria are a threat to the world, causing therapeutic challenges leading to high morbidity and mortality [3].

The first step in solving a problem is to recognize its existence, and one scientific way to recognize the problem of antimicrobial resistance (AMR) at the national level is to conduct nationwide AMR surveillance studies. This could involve the collection of bacterial isolates from different sites across the country and testing their antimicrobial susceptibility to a panel of antibiotics by determining the minimum inhibitory concentration (MIC) of each tested antibiotic against each tested isolate [4]. Resistance rates, as well as MIC_90_ and MIC_50_ (defined as the lowest concentration of the antibiotic at which 90% and 50% of the isolates are inhibited, respectively), are then computed and reported. Evidence has shown that antimicrobial surveillance studies, when used to guide antimicrobial stewardship and infection control, are key in fighting the emergence and spread of AMR at the national and global levels [5,6].

Several antimicrobial susceptibility testing methods have been reported in the literature; however, only one method is recognized as the “gold standard” according to international laboratory standards and is known as broth microdilution (BMD) [7,8,9,10]. This method involves manual testing of cultured bacterial isolates against different, serially diluted antibiotics using microtiter plates and overnight incubation [4]. This is followed by MIC reading and interpretation of the readings according to the most recently published MIC interpretation guidelines by the Clinical and Laboratory Standards Institute (CLSI) [8]. In most cases, antimicrobial testing using BMD is carried out in triplicate (each bacterial isolate is tested three times) to ensure accurate results and eliminate potential outliers. BMD is also accompanied by plating a specific small volume of the bacterial inoculum contained on the microtiter plates in order to ensure that a sufficient number of bacteria is present on the plate as an additional measure to ensure accurate results. Since BMD is highly accurate in determining the MIC of antimicrobial agents against bacteria, it is considered one of two reference methods based on which automated antimicrobial susceptibility testing systems (ASTs) are validated (the second method is known as agar dilution and is more labor and resources intensive) [11]. Therefore, many international scientific groups utilize BMD in conducting AMR surveillance studies, as reported in their published work [12,13,14,15,16,17,18,19,20,21,22,23].

This is the first review to comprehensively discuss the current status of AMR in Saudi Arabia by evaluating rates of antibiotic resistance for common resistant bacteria based on findings from surveillance studies published in the literature. The review also aims to provide general recommendations regarding the importance of national antibiotic resistance surveillance programs, as well as antimicrobial stewardship, infection control, and antibiotic awareness.

## 2. Search Strategy

To search for AMR surveillance studies from Saudi Arabia, a literature search was performed using PubMed and Scopus databases spanning from January 2013 to April 2023. The reason behind the search timeframe is that AMR is a rapidly emerging problem and its prevalence among different bacterial species changes continuously throughout the years and, therefore, including studies dated before 2013 may not be meaningful and relevant to the current AMR situation in the country. The search terms included ‘antimicrobial’, ‘surveillance’, ‘Saudi Arabia’, and the name of the organism in question. Conference proceedings and abstracts were searched separately using the Scopus database. The resulting search items were reviewed carefully to include only pertinent studies that evaluated antimicrobial susceptibility of collected bacterial isolates and to exclude clinical studies.

## 3. Resistant Gram-Positive Pathogens

### 3.1. Methicillin- and Vancomycin-Resistant Staphylococcus aureus

Methicillin-resistant *Staphylococcus aureus* (MRSA) has been a common antibiotic-resistant organism responsible for nosocomial and community-acquired infections since the 1960s [1]. MRSA is the most frequently encountered organism resistant to a single antimicrobial and it was found to be associated with a high economic burden due to medical and societal costs of prolonged hospitalization, mortality, and productivity losses [1,24].

When the literature was searched for surveillance studies involving MRSA in Saudi Arabia, 17 studies were found [25,26,27,28,29,30,31,32,33,34,35,36,37,38,39,40,41]. The number of reported isolates ranged from 4 to 4853 MRSA isolates [25,26,28,29,30,31,32,33,37,38,39,40,41]. The average prevalence of MRSA among all *S. aureus* isolates in the studies that reported it was 32.5% [26,28,31,32,36,38,41], while the prevalence of *S. aureus* among total Gram-positive bacteria was about 24.8% [29,31]. Alam et al. conducted a surveillance study in western Saudi Arabia and revealed that MRSA accounted for over 40% of *S. aureus* isolates [28]. Khan et al. conducted patient, staff, and hospital environmental screening for common nosocomial infections and found that about 50% of MRSA isolates were detected among hospital staff and patients. These findings highlight the significance of implementing surveillance and effective infection control strategies in hospitals and other healthcare facilities. Additionally, Al Musawi et al. conducted another surveillance study that revealed a 10% increase in MRSA resistance over time [30]. Given that linezolid is the last resort for the treatment of MRSA (especially if resistant to vancomycin), resistance to this antibiotic could have a significant effect on the choice of therapy and patient outcomes [42]. Fortunately, four studies from Saudi Arabia found that MRSA isolates had a high susceptibility to linezolid (>90%) [29,30,33,41], whereas two studies found that MRSA had enhanced resistance to linezolid, with susceptibility rates of 55% and 67% [25,32]. Summary of MRSA resistance data is shown in Table 1 and the change in prevalence over the years is displayed in Figure 1.

From the previously mentioned studies, only 6 out of 17 were multicentered [25,34,36,39,40,41]. Automated AST systems were used in most of the studies. VITEK was the most commonly utilized system, which was solely used in five studies [28,30,33,37,38]. MicroScan was also utilized in two studies [27,29], while three other studies reported using multiple automated AST methods within their facility [25,39,41]. The disk diffusion method was used in two studies [31,32]. The remaining studies did not report the utilized method for susceptibility testing [34,36,40]. Regarding the use of molecular testing, only one study reported using the polymerase chain reaction (PCR) method to confirm methicillin resistance through the detection of the *mec*A gene [34].

### 3.2. Vancomycin-Resistant Enterococcus *spp.*

A dozen species belong to the *Enterococcus* genus; however, only two are primarily associated with human infections, *Enterococcus faecium* (77% of clinical isolates) and *Enterococcus faecalis* (9% of clinical isolates) [1]. *Enterococcus* spp. mainly cause healthcare-related infections including urinary tract, intra-abdominal, skin and soft tissue, and biliary tract infections, as well as endocarditis and bacteremia [1]. In the US, vancomycin-resistant *Enterococci* (VRE) is recognized as the second most common antimicrobial resistant organism isolated from blood, urine, and skin, preceded only by MRSA [24]. As VRE bacteremia is associated with a two-fold increase in mortality, VRE infections are generally associated with excess treatment costs compared with vancomycin-susceptible *Enterococcus* spp. [29,43].

Of the Saudi studies found in the literature search, seven were relevant to *Enterococcus* spp. [25,26,36,44], and three focused on VRE [33,40,45]. The total number of isolates included in the studies ranged from 42 to 2266 for *Enterococcus* spp. and from 7 to 57 for VRE [25,33,40,44,45]. The prevalence of VRE among all *Enterococcus* spp. isolates was 13% [36], whereas the prevalence of VRE among total Gram-positive bacteria was about 2.5% [26]. Surveillance conducted over a 10-year period revealed that the rising rate of VRE is most likely attributable to the widespread utilization of broad spectrum antimicrobials [40]. In a study involving 378 different strains of *Enterococcus* spp., 17 of them (4.5%) were proven to be VRE by both phenotypic and genotypic testing [45]. The majority of the 17 VRE isolates were found to be *E. faecium* (*n* = 13/18; 76.4%), whereas the remaining four isolates were found to be *E. gallinarum* [45]. The majority of the 13 *E. faecium* isolates harbored the *vanA* or *vanB* gene, as determined by genotyping (*n* = 8/13; 62%) [45]. Farman et al., identified 17 sequence types (STs) of the *E. faecalis* genomes through multi-locus sequence typing (MLST) analysis [44]. STs included two novel STs (ST862 and ST863) [44]. However, ST179 and ST16 from clonal complex 16 (CC16) were the most frequently occurring STs in the Saudi patients [44]. Furthermore, 22 virulence factors were found in the isolates, most of which were linked with endocarditis, colonization, biofilm development, and cell adhesion [44]. Regarding susceptibility to linezolid, most VRE isolates exhibited high susceptibility rates, reaching more than 85% [25,33,44]. A summary of resistance data for both *Enterococcus* spp. and VRE is shown in Table 1. The trend in the prevalence of VRE over the course of 10 years is shown in Figure 1.

Four of the discussed studies were conducted at a single center [26,33,44,45], while the other three were multicentered [25,36,40]. Automated AST systems were used in most of the studies. Two studies reported using VITEK [33,44], while one study reported using MicroScan [45]. Interestingly, two other studies reported using multiple automated testing methods within their facility [25,26], and the last two studies did not report the method used for susceptibility testing [36,40]. None of the studies utilized molecular testing methods for genotyping purposes.

## 4. Resistant Gram-Negative Pathogens

### 4.1. Enterobacterales

In the WHO list, Gram-negative MDR Enterobacterales (such as *Klebsiella pneumoniae*, *Escherichia coli*, and *Proteus mirabilis*) belong to the critical priority pathogens group. These MDR bacteria are a threat to the world, causing therapeutic challenges leading to high morbidity and mortality [2,3]. This part of the review summarizes the overall rates of antimicrobial resistance in Saudi Arabia among several Enterobacterales species, mostly *K. pneumoniae* and *E. coli*. Studies that evaluated extended-spectrum β-lactamases (ESBL) producing Enterobacterales and carbapenem-resistant Enterobacterales (CRE) are discussed in more detail in later sections of the article.

Eight studies conducted in Saudi Arabia identified in the literature evaluated the susceptibility of Enterobacterales species to different classes of antibiotics, mostly β-lactam antibiotics. Three studies exclusively analyzed urine specimens while the remaining studies evaluated a variety of samples, including blood, urine, sputum, and wound. Most analyzed organisms were *K. pneumoniae*, *E. coli*, and *P. mirabilis*.

*E. coli* isolates from urine samples were evaluated in two studies and they showed high resistance to ampicillin, averaging 90.4% [46,47]. The lowest resistance rate was among carbapenems, ranging between 1.6 and 6.4%. In these studies, >70% of *K. pneumoniae* isolates were resistance to ampicillin. In addition, resistance rates of *K. pneumoniae* isolates against carbapenems were slightly higher than *E. coli* (6.5–31.4%). Both studies also showed that *E. coli* and *K. pneumoniae* isolates had >60% resistance rate to at least four of 20–22 tested antimicrobial agents [46,47].Three studies evaluating resistance rates of different clinical isolates classified *K. pneumoniae*, *E. coli*, and *P. mirabilis* isolates as MDR [48,49,50]. A study by Wani et al. identified 21% of *K. pneumoniae,* 20.7% of *E. coli,* and 9% of *P. mirabilis* isolates as having extensive-drug-resistance (XDR) [49]. The average percentages of susceptibility from the studies are shown in Table 2. Two studies did not specify *Klebsiella* nor *Proteus* species and thus their susceptibility data were not included in the table.

Only two out of the eight studies were multicentered. The first study included four hospitals in Hail and the other study was conducted at three hospitals in Qassim [25,48]. Most studies utilized automated AST systems such as VITEK 2, BD Phoenix, and MicroScan. One study utilized the agar diffusion method for susceptibility conformation [48]. Another study used BMD when indicated; however, the authors did not provide further information regarding the number of isolates that were tested nor under what condition this method was utilized [29]. None of the studies in this section utilized genotypic testing.

### 4.2. Extended-Spectrum β-Lactamases Producing Enterobacterales

β-Lactamase enzymes hydrolyze and breakdown β-lactam antimicrobials, hence obliterating their antimicrobial effect [1]. Extended-spectrum cephalosporins (third and fourth generations) retained their activity against pathogens producing such enzymes until Knothe et al. discovered a β-lactamase that hydrolyzes these cephalosporins and was named ESBL [51]. This enzyme also hydrolyzes the monobactam aztreonam [34]. There are several ESBL genes, such as CTX-M, TEM, and SHV. Worldwide, the most widely reported ESBL genes are CTX-M and TEM [52]. ESBLs are predominantly produced by Enterobacterales (mostly *K. pneumoniae*, *K. oxytoca*, and *E. coli*), but were also reported with *Pseudomonas aeruginosa* and *Acinetobacter baumannii* [53]. These challenging pathogens cause infections in the urinary tract, respiratory tract, intra-abdomen, biliary tract, skin and soft tissue, and blood [1]. As is the case with other antimicrobial-resistant organisms, ESBL-producing organisms are associated with poor clinical outcomes, including prolonged hospitalization, increased mortality, and high hospital costs [1].

Twenty-four Saudi studies were identified in the literature that evaluated the susceptibility of members of Enterobacterales to β-lactam antimicrobials, particularly extended-spectrum cephalosporins [28,29,33,39,47,52,54,55,56,57,58,59,60,61,62,63,64,65,66,67,68,69,70,71]. Seventeen studies provided detailed antimicrobial susceptibility results for each tested organism [28,33,52,54,55,56,57,58,59,60,61,62,63,64,65,66,67]. Eight studies evaluated urine samples, whereas the rest of the studies assessed samples from different sites [33,52,55,58,61,66,67,68].

A study evaluating antibiotic susceptibility of isolates collected from intensive care units found that the majority of *K. pneumoniae* isolates were highly resistant to ceftazidime, cefuroxime and cefotaxime (87–96%). The most active agents were amikacin and meropenem with 4% and 8.6% resistance, respectively. The least active agents for *E. coli* isolates were ceftazidime, cefuroxime, and cefotaxime (56%). Ceftriaxone was the most active antibiotic against *E. coli* with a 6% resistance rate. This study also found that 82.6% of *K. pneumoniae* and 100% of *E. coli* isolates were identified as ESBL [69]. A retrospective cohort study evaluating resistance rates over two years at a single center in Riyadh showed a slight increase of MDR of ESBL *K. pneumoniae* isolates (22% to 26%) [70]. Another retrospective study evaluating resistance trends over five years in the central region of Saudi Arabia identified 37.1% and 27.8% of *E. coli* and *Klebsiella* species isolates as ESBL, respectively. In that study, carbapenem resistance was observed in 38.3% of *E. coli* and 44.2% of *Klebsiella* species isolates [39]. Another study found that 98–100% of ESBL *E. coli* and ESBL *K. pneumoniae* isolates were highly susceptible to carbapenems. In addition, 100% of ESBL *E. coli* isolates were susceptible to tigecycline; however, 8% of *K. pneumoniae* were resistant [65]. A study included >6000 ESBL *E. coli* and >10,000 *K. pneumoniae* isolates over six years found that most isolates were susceptible to carbapenems (92.3–99.8%) [60] Seventy-nine *K. pneumoniae* isolates were evaluated in a study that found that 36.7% were molecularly confirmed to produce ESBL. All the 79 isolates were identified as XDR, and tigecycline and colistin were the antibiotics to which most of the isolates were susceptible [59]. In all studies that applied ESBL genotypic testing, the most common strain was CTX-M among all organisms. The average susceptibility percentages of the studies are shown in Table 3. The evolution of ESBL over ten years is demonstrated in Figure 2.

All studies were single centered except for four: one hospital was in Qassim and two hospitals in Riyadh, three hospitals in Riyadh, three hospitals in Najran, and two hospitals in Hail [33,39,56,61]. Regarding the phenotypic detection of ESBL, the majority of the studies solely utilized automated AST systems. Three studies used the E test as a confirmatory method [56,65,69]. In addition to automated AST systems, some studies utilized double disk diffusion method, double disk synergy test, or a combination of both. In one study, BMD was applied to ensure MIC accuracy of some isolates which were not specified [59]. Two studies did not mention the susceptibility method used [63,70]. Only six studies utilized real-time PCR for genotypic testing [52,56,59,64,65,68].

### 4.3. Carbapenem-Resistant Enterobacterales

Since the evolution of ESBL-producing Enterobacterales, the utilization of carbapenems, one of the powerful antibiotic classes, has globally increased tremendously in response to AMR threat [1]. This increased utilization has led to the emergence of carbapenem resistance [1]. Two identified mechanisms by which Enterobacterales express this resistance are reduction of carbapenem permeability via modifying membrane porins and production of carbapenemase enzymes [72]. The most common carbapenemases worldwide are Ambler class A *K. Pneumoniae* carbapenemase (KPC), class B metallo-β-lactamases (VIM, IMP, NDM), and class D (OXA-48) [1]. Nowadays, CRE are considered the biggest bacterial threat in the world due to the limited treatment options available to combat them and the resultant poor clinical outcomes associated with their infections [1].

From the literature search, 17 studies were conducted in Saudi Arabia assessing the prevalence of CRE [39,54,55,56,57,59,63,69,70,73,74,75,76,77,78,79,80], and two of those evaluated urine samples only [55,75]. One study evaluating Gram-negative isolates from the urine found that 55.7%, 35.8%, and 12.4% of *K. pneumoniae*, *P. mirabilis*, and *E. coli* isolates were potential CREs, respectively. More than 55% of *K. pneumoniae* and 31.5% *of P. mirabilis* were resistant to carbapenems. The study also included other organisms, such as *Enterobacter cloacae*, *K. aerogenes*, and *K. oxytoca*, where the vast majority of the isolates were identified as potential CREs [55]. Additionally, blood isolates from one study classified 46% and 4.3% of *K. pneumoniae* and *E. coli* isolates as CRE, respectively [54]. A study of 31 carbapenem non-susceptible *K. pneumoniae* (*n* = 21) and *E. coli* (*n* = 10) isolates utilized genotypic testing. For *K. pneumoniae*, 66.7% expressed the OXA-48 gene, whereas 33.3% expressed the NDM gene. Among *E. coli* isolates, 60% and 40% had NDM and OXA-48 genes, respectively. In addition, all *E. coli* isolates harboring OXA-48 gene were also positive for the CTX-M (ESBL) gene. All *E. coli* and *K. pneumoniae* isolates were susceptible to tigecycline, and all *K. pneumoniae* and *E. coli* isolates were susceptible to colistin except for one isolate from each genus [56]. A study that evaluated PCR test results for CRE detection in 90 Gram-negative isolates found that the majority of CRE isolates were *K. pneumoniae* (78.9%) and *E. coli* (14.4%), where the most commonly reported gene was OXA-48. Among antimicrobials, colistin and tigecycline were the most effective antibiotics [73]. Khan et al. studied 120 Enterobacterales isolates, mostly *K. pneumoniae* (42%) *and E. coli* (41%). Among these isolates, 50% and 4.9% of *K. pneumoniae* and *E. coli* were CRE, respectively. Interestingly, this was the first study in Saudi Arabia to report a triple CRE gene resistance (KPC, NDM, OXA-48), which was found in 17 *K. pneumoniae* isolates. The most effective antibiotics in this study were colistin (susceptibility = 100%) against both organisms, followed by amikacin, with 50% and 100% susceptibility rates against CRE *K. pneumoniae* and CRE *E. coli,* respectively [74]. A retrospective cohort study assessing resistance rate over two years showed a reduction of colistin susceptibility against CRE *E. coli* (97%, 86%) and CRE *K. pneumoniae* (80%, 76%) [70]. However, the study did not report the method by which colistin susceptibility was assessed. Taha et al. included isolates that were phenotypically resistant to carbapenems and utilized a PCR test to identify common CRE genes. Similar to other studies, the most common CRE genes identified in *K. pneumoniae* and *E. coli* were OXA-48, followed by NDM. CRE *E. coli* isolates were highly susceptible to tigecycline and colistin, as reported in most studies. However, *K. pneumoniae* resistance to colistin (19.2%) and tigecycline (21.6%) was higher than reported in similar types of study [78]. A multicenter prospective study that included eight hospitals was conducted to describe molecular features of CRE *E. coli* and CRE *K. pneumoniae* that were isolated from 189 patients. Most patients were from the western region hospitals. The study did not provide data regarding antimicrobial susceptibility data; however, it provided valuable information about the prevalence of CRE in Saudi Arabia based on genotyping results. The two most common genes reported were OXA-48 (*n* = 131) followed by NDM (*n* = 32). It should be noted that some patients had pathogens that harbored more than one CRE gene [79]. A multicentered study was conducted at 13 hospitals in Saudi Arabia to assess the molecular characterization of 519 CRE. Most isolates were collected from the southern region and only six isolates were collected from the eastern region. The two most reported organisms were CRE *K. pneumoniae* (456) and CRE *E. coli* (21) isolates. The most common gene reported was OXA-48 (71.2%) for *K. pneumoniae* and NDM (69%) for CRE *E. coli*. Unfortunately, the study did not report susceptibility nor MIC data due to different phenotypic methods utilized at different sites [77]. The average susceptibility percentages from the studies are shown in Table 4. The emergence and spread of CRE over the years are shown in Figure 2.

Nine out of the 18 studies were multicentered. The study that included the largest isolates (*n* = 29, 393) was conducted at two hospitals in Riyadh and one hospital in Qassim over a five-year period [39]. The second largest multicenter study was conducted at 13 hospitals across the country and included 456 isolates [77]. As for CRE phenotypic testing, five studies utilized automated AST systems as the only method [39,54,55,73,76]. Five studies used the E test in addition to automated AST systems [69,77,78,79,80]. Three studies applied the Modified Hodge test (MHT) which is no longer recommended for CRE production testing according to CLSI [56,59,79]. One study mentioned using BMD on some isolates [59]. In another study, BMD was used to test susceptibility of colistin and tigecycline [75]. It is important to note that some studies have shown that the E. test may fail to detect all OXA producers since some OXA isolates show susceptibility or low resistance toward carbapenems [80]. Ten studies utilized PCR for genotypic testing [56,59,73,74,75,76,77,78,79,80]. In spite of the era of genotypic testing, current assays contain a limited number of genes they can capture, and organisms may exhibit other resistance mechanisms that are not detected with the current susceptibility testing methods. [77].

### 4.4. Pseudomonas aeruginosa

*P. aeruginosa* is one of the leading Gram-negative pathogens responsible for healthcare-associated infections worldwide, including pneumonia, urinary tract infections, bloodstream infections, and surgical site infections [1]. *P. aeruginosa* is uniquely troublesome in that it is inherently resistant to many antimicrobial classes and is able to develop resistance upon exposure to antimicrobials [81]. Mechanisms of drug resistance in *P. aeruginosa* vary from enzyme production (ESBL or carbapenemases) to blockage of antibiotic entry or efflux of antibiotics after successful cellular entry [1]. The threat to healthcare is particularly associated with MDR strains (i.e., resistant to three or more drugs from different antimicrobial classes) of *P. aeruginosa* [1]. Therefore, infections caused by MDR *P. aeruginosa* are often associated with high morbidity and mortality, as well as longer hospital length of stay, which ultimately result in an economic burden [1].

Various surveillance studies reported different resistance rates of *P. aeruginosa.* A total of 11 studies were identified that evaluated the susceptibility of *P. aeruginosa* isolates [29,33,39,48,50,63,66,82,83,84,85], while two studies specifically evaluated MDR strains of *P. aeruginosa* [83,84], and another two studies did not specify the species of the evaluated *Pseudomonas* isolates [39,66]. One prospective study from a small hospital identified the antibiotic susceptibility of 52 *P. aeruginosa* isolates from patients in medical and non-medical wards. The results showed that *P. aeruginosa* was found mostly in wound swabs and had variable susceptibility profiles to antibiotics, where the highest susceptibility rate was reported with piperacillin/tazobactam at 61.5%, followed by amikacin and imipenem with 50% with each. The remaining antibiotics from the other classes (antipseudomonal β-lactams, fluoroquinolones, and aminoglycosides) had activities ranging from 12.5 to 27% [29]. Over a 5-year period, a multicenter surveillance study evaluated antibiotic resistance patterns in uro-pathogens, and found that *P. aeruginosa* was the third organism responsible for urinary tract infections, mostly in females, and its prevalence increases with age. A total of 66% resistant strains of *P. aeruginosa* were noticed in 2017, but by 2019 that number had dropped to 25%. Since 2015, gentamicin has shown reduced activity against *P. aeruginosa* isolates between 2015 and 2017 with resistance rates ranging between 25–66%, followed by improved activity in 2019 with 100% susceptible isolates [33]. A longitudinal surveillance study of antibiotic susceptibility patterns among Gram-negative bacteria was conducted by evaluating the institutional antibiogram over a period of six years. As the susceptibility of *P. aeruginosa* isolates decreased with time over five years (2013–2018), it remained stable to ceftazidime and meropenem. However, it showed slightly improved susceptibility to cefepime, amikacin, ciprofloxacin, and gentamicin [39]. Two retrospective studies determined the antimicrobial resistance rates of different Gram-negative bacteria in intensive care units (ICUs), In one study, *P. aeruginosa* isolates (*n* = 69) were mostly recovered from urine (36.7%), blood (32.1%), and sputum (24.5%), and showed high resistance to piperacillin/tazobactam, cephalosporins, meropenem, and aztreonam, with resistance rates of 46.3 to 53.3%, but moderate resistance to aminoglycosides (18.8–31.7%) [50]. Similar findings were reported from a recent study (between 2019 and 2022) of 1024 isolates, where 221 (21.5%) isolates were reported as MDR, mostly recovered from wound infections, followed by respiratory infections. Total resistance rates over the study period were the highest against carbapenems (61–77.8%), whereas the lowest rates were against ciprofloxacin (11.3%), amikacin (15.3%), and tobramycin (18%). Resistance to colistin was reported at 2.4%; however, since the automated AST was used to assess the susceptibility, such value may not be reliable [83]. Furthermore, in a matched case-control study, respiratory specimens were the main source of 90 *P. aeruginosa* isolates that were mostly susceptible to aminoglycosides (86.7%) and cefepime (84.4%), moderately susceptible to piperacillin/tazobactam, ceftazidime, and ciprofloxacin (71.1–77.8%), and least susceptible to carbapenems (58.9%) [82]. A prospective multicenter study from the western region investigated the susceptibility rates of 121 *P. aeruginosa* isolates to various antibiotics. Respiratory tract was the major source among all specimen sources. The isolates were reported to have low to moderate resistance to antipseudomonal antibiotics, where the highest resistance rate was reported against meropenem (30.6%) and imipenem (19%). Resistance rates were the lowest against piperacillin/tazobactam (4.9%), followed by amikacin (7.4%), and cefepime (8.3%). MDR rate was 10.7% [85]. A retrospective study evaluated the clinical and microbiological characteristics of MDR *P. aeruginosa*. Respiratory samples were the most common sources of MDR and XDR isolates. The MDR and XDR isolates had the highest resistance rates against anti-pseudomonal cephalosporins, carbapenems, and aztreonam, with susceptibility rates ranging between 24.3 and 29.5%, though, low rates of resistance were reported against colistin and amikacin with susceptibility rates above 90%. All XDR *P. aeruginosa* isolates were resistant to ciprofloxacin and levofloxacin [84].

Two studies evaluated the resistance pattern of *Pseudomonas* isolates without identifying the species. The first study assessed the antibiotic resistance in uro-pathogens, where *Pseudomonas* spp. were the third most common Gram-negative organisms causing the infection (9.2%) after *E. coli* and *Klebsiella* spp. In this study, *Pseudomonas* spp. exhibited high susceptibility rates to amikacin (80%). However, only 62.3%, 62.3% 56.7%, and 55.3% were susceptible to gentamicin, meropenem, ciprofloxacin, and piperacillin/tazobactam, respectively [66]. The second study was a multicenter study of healthcare-associated infections assessing the change in resistance pattern over five years. In this study, *Pseudomonas* spp. ranked as the fourth most causative pathogen, particularly in the ICU. Resistance rates against all antibiotics fluctuated over the years with the lowest resistance rates being observed against aminoglycosides followed by carbapenems [39]. It should be noted, however, that the lack of specification of the *Pseudomonas* species in these studies may have impacted the distribution of the susceptibility rates to different antibiotics. Table 5 lists the average percentages of susceptibility, while Figure 3 shows the change in the prevalence of MDR *P. aeruginosa* isolates over ten years.

Most of the studies that evaluated the susceptibility of *P. aeruginosa* utilized automated AST systems. The most commonly used system, VITEK, was used in four studies [33,66,82,83]. The second most frequently used method, noted in three studies, was MicroScan [29,63,85]. Two studies employed BD Phoenix [50,84], while two others reported using numerous automated AST techniques at their institutions [39,48]. None of the included studies utilized molecular testing for genotypic identification except one study by Said et al. who used GeneXpert [48].

### 4.5. Acinetobacter baumannii

*A. baumannii* is the major pathogen responsible for the majority of infections and hospital outbreaks among identified members of the *Acinetobacter* genus [86]. Most of these infections are caused by MDR *A. baumannii* strains [1]. Notably, several reports have demonstrated the spread of MDR *A. baumannii* worldwide [87]. Similar to MDR *P. aeruginosa*, MDR *A. baumannii* is also linked to poor prognostic outcomes as well as increased hospital costs [1]. Furthermore, XDR *A. baumannii* has also been reported in several reports where these strains exhibit resistance to all but one or two antibiotic classes [88]. XDR *A. baumannii* are extremely difficult to treat [89], thus the identification of their occurrence, as well as the occurrence of MDR strains, is of a prime importance in helping to guide infection control and antimicrobial stewardship efforts.

The literature search showed nine studies involving *A. baumannii* that were conducted in Saudi Arabia [28,48,63,90,91,92,93,94,95]. The majority of the isolates were from respiratory and blood samples. A multicentered study conducted at four hospitals in Hail evaluated resistance among Gram-negative bacteria, which included 82 *A. baumannii* isolates, which were all classified as pandrug-resistant (PDR). Isolates had >65% resistance rate to most tested antibiotics, including tigecycline (69.2%). The most active agent was colistin with a reported susceptibility rate of 94.9% [48]. A study evaluated antimicrobial susceptibility of 135 *A. baumannii* found that 58.5% of isolates were MDR and none were PDR. A total of 75 isolates were resistant to meropenem and imipenem. Among the tested antibiotics, colistin, minocycline, and tigecycline were the most active agents with reported susceptibility rates of 100%, 72%, and 67%, respectively. Most isolates carried OXA-type genes. In addition, 84%, 18.5%, and 1.5% of isolates carried IMP, VIM, and NDM, respectively [91]. In another study, 82 *A. baumannii* isolates were collected from respiratory samples in 2019 (*n* = 35) and in 2020 (*n* = 47) were mostly identified as XDR. Only one isolate in 2019 (1.2%) was susceptible to tigecycline in comparison to seven isolates in 2020 (14.8%). The most active agent was colistin with a reported susceptibility rate of 91.9–100%. Carbapenemase enzymes were detected in 50 (61%) isolates, including VIM-2 (*n* = 21), IMP-2 (*n* = 19), and NDM-1 (*n* = 10) [92]. A retrospective study assessed *A. baumannii* isolates that were collected from 115 respiratory samples and found that 62%, 35%, and 3% of isolates were MDR, XDR, and PDR, respectively. Isolates were highly resistant to almost all tested antibiotics, including carbapenems (98%). In addition, 15% of isolates were resistant to colistin and 3% to tigecycline [93]. A multicentered study was conducted at five hospitals in Riyadh, Jeddah, Alhsa, Madinah, and Dammam to assess the prevalence of carbapenem resistant Gram-negative bacteria from blood samples. The study identified 496 (71.2%) carbapenem-resistant *A. baumannii* (CRAB) of all *A. baumannii* isolates. It is important to note that most of the samples were collected from Riyadh and Jeddah [94]. A retrospective study included 103 MDR *A. baumannii* isolates that were 98.1% and 57.3% resistant to imipenem and meropenem, respectively, whereas the lowest resistance rate was against tigecycline (5.9%). The most commonly identified genes were OXA-51 (89.3%), VIM (88.3%), and NDM (84.5%) [95]. The average percentages of susceptibility are presented in Table 5. Figure 3 tracks the trend of MDR *A. baumannii* prevalence.

Two out of the nine studies were multicentered and the largest study included 115 *A. baumannii* isolates [94]. Seven studies utilized automated AST systems for susceptibility testing [28,63,93,95] and one study did not mention what method was used. One study used BMD to confirm susceptibility of isolates to colistin [91]. Two studies applied the Kirby Bauer Disk Diffusion method [92,93], and one study used disk diffusion and agar diffusion methods, E test, and checkerboard assay [90]. One study utilized MHT to test for carbapenemase production which, as mentioned earlier, is no longer recommended by CLSI [92]. Only three studies utilized PCR for genotypic testing [91,92,95].

## 5. Future Directions

The literature search that has been carried in this study examining the last decade’s AMR surveillance in Saudi Arabia yielded numerous studies, most of which were either reported from a single center, included only certain bacterial species, and/or did not utilize BMD as the antimicrobial susceptibility testing method. This finding indicates the need to conduct large nationwide surveillance studies based on international standards. While genotyping is helpful in identifying genes associated with antibiotic resistance, phenotyping remains the most common method to test bacterial susceptibility to antibiotics since not all clinical microbiology laboratories may have the sufficient resources to perform genotype tests. To overcome this issue, some phenotyping-based algorithms were developed to identify certain resistance mechanisms. One recent example is the MIC-based algorithm to identify *P. aeruginosa* isolates that exhibit carbapenem resistance via carbapenemase production, since this bacterium may exhibit carbapenem resistance using different mechanisms [96]. Such an algorithm can be used by microbiology labs to help guide therapy by suggesting which isolates can be potentially treated with carbapenemase inhibitor combinations. Another limitation of this review was the gaps in reporting antimicrobial resistance rates, when some years had multiple studies while other years included very few studies as shown in the figures (the number of studies on each year point), not to mention the variation in the strains of bacteria reported in these studies. Therefore, the average percentages of susceptibility to antimicrobials were calculated to represent the mean of the values of all the studies. The figures were also created to track the change in resistance development over the course of ten years.

Saudi Arabia has developed a remarkable vision for the year 2030 to be a pioneer in numerous developmental areas, including health, economy, industry, and environment to name only a few [97]. The sectors that would most benefit from such AMR surveillance program are the health, economic, and education sectors. Having such a program could have the potential to contribute to the 2030 vision in several ways. One of the vision’s themes is *Vibrant Society with Strong Foundations* where one of this theme’s focuses is *Caring for Our Health* which envisions increasing the life expectancy of Saudis and optimizing healthcare quality. In order to fulfill this vision, a focus on reducing infectious diseases and providing preventive care is highly encouraged. A national antimicrobial resistance surveillance program would significantly contribute to this vision, since the first step in correcting a problem and preventing it is recognizing it and realizing its magnitude, followed by strategic planning and provision of specific recommendations on tackling it with subsequent action. Comprehensively evaluating the rates of AMR at the national level through the collection of hundreds of bacterial isolates and testing their antimicrobial susceptibility to commonly used antibiotics using the most accurate method approved by international laboratory standards should provide current adequate knowledge on the nature and extent of the AMR problem in the country. Once this becomes known, it should be immediately followed by strategic planning while providing pertinent recommendations involving the encouragement of antimicrobial stewardship and the improvement of overall clinical practice when it comes to the treatment of infectious diseases, as well as infection control and prevention in healthcare facilities. A recent review by a Saudi infectious diseases pharmacy group provided a comprehensive guidance on how to utilize the expertise of infectious diseases specialized pharmacists in fulfilling the 2030 vision by improving healthcare through implementation of antimicrobial stewardship and spreading awareness regarding antibiotic use [98]. Having national antimicrobial resistance data can support such efforts in achieving their goals.

Beside improving healthcare outcomes, and based on evidence from the literature, applying these recommendations should result in improving the economy through big savings in hospital costs, as well as other indirect costs, such as avoidance of work loss. Therefore, the proposed surveillance program could contribute to an additional 2030 vision theme titled *Thriving Economy Investing for the long-term*.

The third and last vision theme to which a national antimicrobial resistance surveillance can potentially add is *An Ambitious Nation Responsibly Enabled* where it focuses on *Being Responsible for Our Lives* and *Being Responsible to Society*. By providing accurate and reliable information on AMR rates in Saudi Arabia and publicly sharing this information and its interpretation with the Saudi public, the surveillance program could contribute to society by educating it about an important global problem and encouraging it to discuss this with healthcare providers, colleagues, family, and friends in order to spread awareness and promote appropriate AMR prevention actions at the public levels, such as avoiding antibiotic misuse and premature discontinuation of antibiotic treatment courses.

## 6. Conclusions

Pathogens with AMR are associated with significant negative clinical and economic impacts on patients, healthcare systems, and governments. MDR Gram-positive and Gram-negative organisms are prevalent in Saudi Arabia. Globally, AMR is increasing in prevalence and novel resistance mechanisms are continuously evolving. Reputed scientific groups that conducted and published surveillance studies evaluating AMR consistently utilized BMD as the microbiological method for antimicrobial susceptibility testing, since it is the method considered “gold standard” according to international laboratory standards. These surveillance studies also included isolates collected from multiple healthcare facilities to provide robust antimicrobial susceptibility data at the national and global levels. While a total of 76 studies from Saudi Arabia were identified in the literature from the last decade that evaluated antimicrobial susceptibility and AMR rates of various bacterial strains, most were limited to a single center and 71 (93.4%) did not utilize BMD to test antimicrobial susceptibility. Additionally, some studies used BMD but only for some of the isolates or to test the susceptibility to colistin. Because many of these studies assessed AMR at different points of time throughout the years of the last decade, it would be difficult to draw specific conclusions on national AMR rates for a specific period of time. Therefore, a large national multicenter surveillance antimicrobial susceptibility study (or series of studies) involving the commonly known resistant organisms discussed in this review against a panel of the commonly used antibiotics available in the country that are approved by the Saudi Food and Drug Authority is indicated. Using the most accurate antimicrobial susceptibility testing method, BMD, is also highly indicated. Such a program is important to fulfill this gap in knowledge and to contribute to global society by providing recent AMR data from Saudi Arabia to the World Health Organization’s program, GLASS, as well as to the ResistanceMap project by the Center for Disease Dynamics, Economy and Policy [99,100].

## Figures and Tables

**Figure 1 microorganisms-11-02086-f001:**
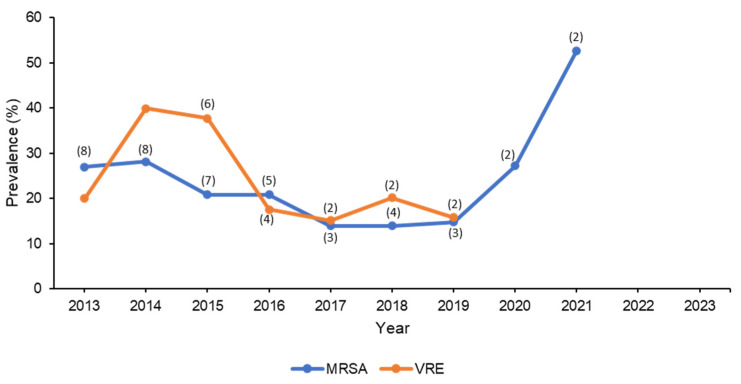
Trend in the change in prevalence of MRSA and VRE over 10 years. Note: No studies reported prevalence after 2021. The numbers in between parentheses indicate the number of studies from which the mean prevalence was calculated. Studies that reported overall prevalence over a period of several years had the average prevalence calculated and typed in the middle year of that period.

**Figure 2 microorganisms-11-02086-f002:**
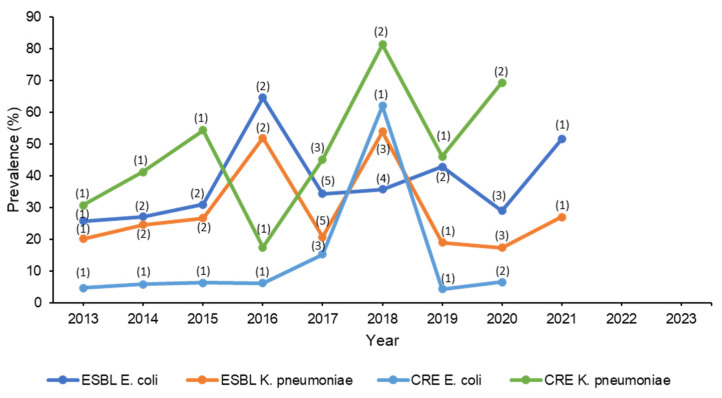
Trend in the change in prevalence of resistant Gram-negative bacteria over ten years. Note: No studies reported prevalence after 2021. The numbers in between parentheses indicate the number of studies from which the mean prevalence was calculated. Studies that reported overall prevalence over a period of several years had the average prevalence calculated and typed in the middle year of that period.

**Figure 3 microorganisms-11-02086-f003:**
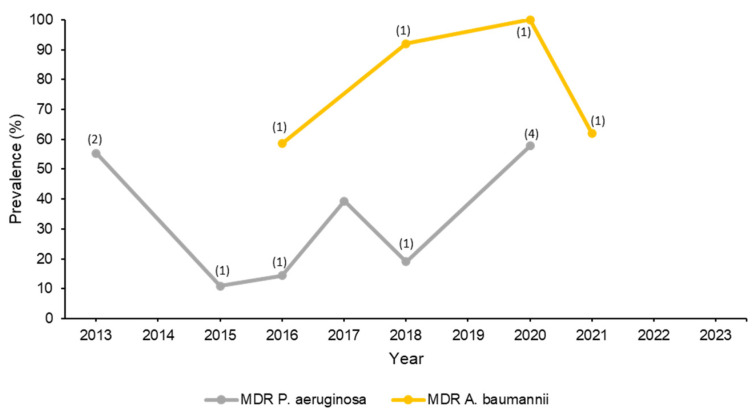
Trend in change in prevalence of MDR *Pseudomonas aeruginosa* and *Acinetobacter baumannii* over ten years. Note: No studies reported prevalence after 2021. The numbers in between parentheses indicate the number of studies from which the mean prevalence was calculated. Studies that reported overall prevalence over a period of several years had the average prevalence calculated and typed in the middle year of that period. Also, 3 of 11 *P. aeruginosa* studies and 5 of 9 *A. baumannii* studies did not report percentages of MDR isolates. Studies that only reported MDR *A. baumannii* (without mentioning their proportion out of all isolated *A. baumannii* within the institution) were not included in the figure, since their prevalence is unknown.

**Table 1 microorganisms-11-02086-t001:** Average percentage of susceptibility of Gram-positive organisms to antibiotics in Saudi Arabia between 2013–2023.

Antibiotic	*Staphylococcus aureus* (*n* = 5650)	MRSA (*n* = 3097)	VRE (*n* = 24)	*Enterococcus* Species * (*n* = 2266)	*Enterococcus* *faecalis* (*n* = 149)
Amoxicillin/clavulanic acid	72.1	16.9	-	-	-
Amikacin	-	23	-	-	-
Ampicillin	47.4	5.2	100	83	95
Azithromycin	50	-	-	-	-
Cefazolin	100	100	-	-	-
Ceftriaxone	66.7	-	-	-	-
Cefotaxime	-	14.4	-	-	-
Cefoxitin	56.9	11.5 **	100	29.7	-
Chloramphenicol	54.6	-	-	-	-
Ciprofloxacin	74	58.6	-	35.9	67
Clindamycin	88.9	62.4	28.6	27.9	5
Cloxacillin	82.9	25.6	-	82.6	-
Daptomycin	100	97.9	-	-	-
Doxycycline	53.4	51	-	17.7	-
Erythromycin	73.9	52.6	-	28.8	9
Fosfomycin	90.5	100	-	-	-
Fusidic Acid	94.1	58.4	-	-	-
Gentamicin	86.2	59.9	-	37.4	50
Imipenem	36.8	18.2	-	-	-
Levofloxacin	76.5	49.3	71.4	64	55
Linezolid	80.5	86.8	87.9	97.3	97
Moxifloxacin	83	44.9	85.7	-	55
Mupirocin	80.9	96.6	-	-	-
Nitrofurantoin	84.7	68.1	43	80.9	95
Oxacillin	78.3	4.3 **	-	-	-
Penicillin	50.1	0.3	85.7	-	85
Quinupristin	100	-	-	-	5
Rifampin	94.1	84	-	-	-
Streptomycin	-	-	-	-	70
Quinupristin/dalfopristin	94.1	-	-	-	-
Teicoplanin	97.1	91	38.5	-	-
Tigecycline	86.2	62.5	100		100
Tetracycline	75.2	69.2	-	35.3	85
Tobramycin		54.5	-	-	-
Trimethoprim/sulfamethoxazole	82.6	55.6	28.6	67.4	-
Vancomycin	87.7	86.9	0	93	97

MRSA, methicillin-resistant *Staphylococcus aureus*; VRE, vancomycin-resistant *Enterococci.* * Includes both *E. faecalis* and *E. faecium.* ** This average percentage (4.3%) of oxacillin susceptibility was calculated from three different studies, two of which reported 0% susceptibility and one reported 12.8% susceptibility [32,38,41]. Similarly, the cefoxitin average susceptibility percentage (11.5%) was calculated from three different studies that reported susceptibilities of 21%, 0%, and 13.3%, respectively [25,30,41].

**Table 2 microorganisms-11-02086-t002:** Average percentage of susceptibility of Enterobacterales to antibiotics in Saudi Arabia between 2013–2023.

Antibiotic	*Escherichia coli* (*n* = 50,589)	*Klebsiella pneumoniae* (*n* = 18,539)	*Klebsiella aerogenes* (*n* = 858)	*Klebsiella oxytoca* (*n* = 739)	*Proteus mirabilis* (*n* = 5604)	*Providencia stuartii* (*n* = 1733)	*Morganella morganii* (*n* = 1754)	*Enterobacter cloacae* (*n* = 1363)	*Citrobacter freundii* (*n* = 596)
Amikacin	82.5	81.3	30	94.7	58	76.6	73	72.3	88.1
Amoxicillin/ clavulanic acid	58.8	52.5	12.2	33.3	49.9	19.6	24.4	40	9.1
Ampicillin	33.4	17.3	6.5	5.9	25.3	16.1	4.9	27	0
Cefepime	62.3	59.4	9.8	60.7	38.5	38.4	14.5	47.9	45.7
Cefoxitin	77.2	69.4	53.5	66.7	71.4	73.1	73.1	28	41.5
Ceftazidime	65.2	57.9	12.5	26	36.5	37.3	12.8	57.3	37.6
Ceftriaxone	50.9	44.1	-	-	20.2	54.5	-	-	-
Cefuroxime	46.4	43.8	-	-	27.4	27.3	0	45.5	-
Ciprofloxacin	58.5	59.9	11.1	11.9	35.6	18.7	38.4	70.2	54.2
Colistin	79.3	83.7	53.8	71.4	1.6	1.5	0	60.9	93.3
Ertapenem	82.4	65.7	66.7	100	72.8	77.2	-	87.5	85.7
Gentamicin	74.2	69.4	28	58.2	39.1	21.7	31.3	70.3	58.6
Imipenem	88.5	75.6	94.2	98.1	67.2	54.9	95.6	95.9	71
Levofloxacin	51.9	60.3	20.5	28.9	20	19.5	13.1	55.6	37.9
Meropenem	89.1	73.8	100	100	82.2	73.1	94.1	50	42.9
Nitrofurantoin	77.5	44.2	33.3	38.7	19.7	7	3	61	63.1
Piperacillin/ tazobactam	77.7	64.4	58.5	70.8	70.8	72	53.3	75.2	65.2
Tigecycline	85.4	76.4	91.7	100	0	48.4	50	82.6	91.7
Trimethoprim/ sulfamethoxazole	51.5	54.7	40	23.1	25.9	33	38.7	52.6	29.3

**Table 3 microorganisms-11-02086-t003:** Average percentage of susceptibility of extended-spectrum β-lactamase producing Enterobacterales to antibiotics in Saudi Arabia between 2013–2023.

Antibiotic	*Escherichia coli* (Suspected ESBL) (*n* = 152)	*Escherichia coli* (Confirmed ESBL) (*n* = 7277)	*Klebsiella pneumoniae* (Confirmed ESBL) (*n* = 1382)
Amikacin	87.2	84.6	83.2
Amoxicillin	0	-	-
Amoxicillin/clavulanic acid	38.7	39.7	41.6
Ampicillin	6	8.8	16.1
Aztreonam	17.3	5.2	0
Cefepime	30.2	28.7	33.3
Cefoxitin	61.5	78.2	87.7
Ceftazidime	26.3	14.5	9.6
Ceftriaxone	25.7	22.7	-
Cefuroxime	3.7	-	-
Ciprofloxacin	23.6	32.5	49.1
Gentamicin	51.3	60.1	46.2
Imipenem	91.7	90.4	97.9
Levofloxacin	17.4	-	-
Meropenem	98	88	99.1
Nitrofurantoin	66	79.1	38.4
Piperacillin	0	-	-
Piperacillin/tazobactam	70.9	68.2	71.3
Tigecycline	100	75.3	72
Trimethoprim/sulfamethoxazole	22.6	39.9	36.1

ESBL, extended-spectrum β-lactamase.

**Table 4 microorganisms-11-02086-t004:** Average percentage of susceptibility of carbapenem-resistant Enterobacterales to antibiotics in Saudi Arabia between 2013–2023.

Antibiotic	*Escherichia coli* (Suspected CRE) (*n* = 41)	*Escherichia coli* (Confirmed CRE) (*n* = 106)	*Klebsiella pneumoniae* (Suspected CRE) (*n* = 42)	*Klebsiella pneumoniae* (Confirmed CRE) (*n* = 337)	*Klebsiella pneumoniae* (Suspected CRE/ESBL) (*n* = 78)
Amikacin	100	62.1	71.4	52.4	42.3
Amoxicillin/clavulanic acid	-	0	-	0	9
Cefepime	60.9	2	26.2	0.33	24.4
Cefotaxime	48.8	25	21.4	0	-
Ceftazidime	68.3	1.6	19	3.8	8.7
Ceftriaxone	48.8	4	21.4	0.5	26
Ciprofloxacin	-	29.5	-	3.9	23
Colistin	100	97.5	100	92	-
Gentamicin	53.7	44.3	52.3	25.8	55
Imipenem	95.1	21.4	50	11.1	75.6
Meropenem	95.1	28.3	50	11.8	70.5
Piperacillin/tazobactam	-	0	-	0	24.4
Tigecycline	-	92.2	-	89.8	-
Trimethoprim/sulfamethoxazole	-	52.1	-	7	29

CRE, carbapenem-resistant Enterobacterales.

**Table 5 microorganisms-11-02086-t005:** Average percentage of susceptibility of *Pseudomonas aeruginosa* and *Acinetobacter baumanii* to antibiotics in Saudi Arabia between 2013 and 2023.

Antibiotic	*Pseudomonas aeruginosa* (*n* = 8995)	*Acinetobacter baumannii* * (*n* = 1082)
Amikacin	63.85	36.3
Aztreonam	48.45	-
Cefepime	52.1	-
Ceftazidime	50.4	-
Ciprofloxacin	54.3	1.25
Colistin	79.25	97.5
Gentamicin	51.79	16
Imipenem	50.7	17.6
Levofloxacin	40.2	2.8
Meropenem	32.84	14.4
Piperacillin/tazobactam	60.3	-
Tigecycline	-	21.4
Tobramycin	80	-

Studies that did not specify the species of *Pseudomonas* and *Acinetobacter* were not included in this table. * Since all the included studies evaluated multidrug-resistant isolates, except one [28], the averages presented were from those studies.

## Data Availability

Data are available form the authors upon request.

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
