# Peer review of "An Overview of Antimicrobial Resistance in Saudi Arabia (2013–2023) and the Need for National Surveillance"

_microorganisms, 2023, doi:10.3390/microorganisms11082086_

Round 1

Reviewer 1 Report

The manuscript entitled “An overview of antimicrobial resistance in Saudi Arabia (2013-2023) and the need for national surveillance” describes a review on antimicrobial resistance in Saudi Arabia in the last ten years. Overall, this manuscript provides some interesting information about the antimicrobial resistance of the most important gram positive ang gram negative bacteria that isolated in several studies that performed in Saudi Arabia.

According to this manuscript, the following comments should be addressed before considering its publication:

General comments:

The tables of the manuscript display unclear data. Add the number of total isolates in each table.

Authors should highlight the incidence of multi drug resistant isolates in each study used in this review

Specific comments.

MRSA isolates according to CLSI is considered resistant to oxacillin and/or cefoxitin, While in Table 1. 4.3 and 11.5 % of MRSA isolates were resistant to oxacillin and cefoxitin, respectively. Please revise the table and clear this point.  Also, 23.6 % of VRE isolates are resistant to vancomycin?? Revise and fix please.

Authors should mention if these studies used molecular techniques to detect the antimicrobial resistance genes.

Authors should focus on Linzolid resistance as it is the last drug of choice for treatment of MRSA, VRSA and VRE.

Line 407: Correct the title into: Acinetobacter baumannii.

Author Response

Dear reviewer,

We would like to thank you for taking the time to review and evaluate our manuscript. We have addressed all your comments and have revised the manuscript accordingly. An updated version of the manuscript highlighting the changes is attached. Please find below the responses to your comments.

  1. The tables of the manuscript display unclear data. Add the number of total isolates in each table.

Response: The total number of isolates was added to the title of each column of all the five tables in the manuscript. Also, to address this comment, as well as the comment of reviewer 2, we created figures showing the trend of evolution of resistance over the years for the major resistance patterns (i.e., MRSA, VRE, ESBL, CRE, MDR Pseudomonas and MDR Acinetobacter). The figures demonstrate the average percentage of prevalence that was reported in the studies of each year.

  1. Authors should highlight the incidence of multi drug resistant isolates in each study used in this review.

Response: The included studies in our review did not report incidence per year, but rather reported prevalence within a certain time frame, which we reported and summarized in our manuscript. Additionally, we specifically reported the prevalence of certain drug-resistance patterns, such as MRSA, VRE, ESBL and CRE. However, to address this comment, rates of MDR isolates of Pseudomonas aeruginosa and Acinetobacter baumannii are now displayed in figure 3 (page 12).

  1. MRSA isolates according to CLSI is considered resistant to oxacillin and/or cefoxitin, While in Table 1. 4.3 and 11.5 % of MRSA isolates were resistant to oxacillin and cefoxitin, respectively. Please revise the table and clear this point. Also, 23.6 % of VRE isolates are resistant to vancomycin?? Revise and fix please.

Response: These were all revised and fixed. We apologize for the mistake.

  1. Authors should mention if these studies used molecular techniques to detect the antimicrobial resistance genes.

Response: The use of molecular techniques was added to the parts that lacked them, which were MRSA (lines 127-129) and Pseudomonas aeruginosa (lines 447-449). None of the VRE studies reported using molecular method as stated on lines 167-168. However, it was already mentioned in the other parts (i.e., Enterobacterales, ESBL, CRE, and Acinetobacter baumannii).

  1. Authors should focus on Linezolid resistance as it is the last drug of choice for treatment of MRSA, VRSA and VRE.

Response: Thank you for the suggestion. Statements on susceptibility to linezolid were added to lines 103-108 and 158-159.

  1. Line 407: Correct the title into: Acinetobacter baumannii.

Response: Corrected.

Reviewer 2 Report

The current and highly impactful topic of antibiotic resistance is a constant concern, and antibiotic stewardship policies represent a global issue. As antibiotic resistance has increased in recent years, especially in the post-pandemic period, I don't believe that these average resistance tables provide a clear picture of what is happening.

It would have been interesting if the evolution of this resistance was being tracked for each pathogen, identifying the causes and suggesting solutions to improve antibiotic administration.

Author Response

Dear reviewer,

We would like to thank you for taking the time to review and evaluate our manuscript. We have addressed all your comments and have revised the manuscript accordingly. An updated version of the manuscript highlighting the changes is attached. Please find below the responses to your comments.

  1. The current and highly impactful topic of antibiotic resistance is a constant concern, and antibiotic stewardship policies represent a global issue. As antibiotic resistance has increased in recent years, especially in the post-pandemic period, I don't believe that these average resistance tables provide a clear picture of what is happening.

It would have been interesting if the evolution of this resistance was being tracked for each pathogen, identifying the causes and suggesting solutions to improve antibiotic administration.

Response: To address this comment, we created three figures showing the trend of evolution of resistance throughout the ten years for the major resistance patterns (i.e., MRSA, VRE, ESBL, CRE, MDR Pseudomonas and MDR Acinetobacter). The figures demonstrate the average percentage of prevalence that was reported in the studies of each year. The figures appear on pages 4 (figure 1), 8 (figure 2), and 12 (figure 3).

Round 2

Reviewer 1 Report

Table 1. regarding methicillin and vancomycin resistant isolates, authors replaced the old number with zero; which is not correct. Under VRE heading in Table 1, the percent of vancomycin resistance should be 100%, how it could be 23.6 and later replaced with zero????

The same for MRSA. Authors should revise all the cited numbers in this table.

Author Response

Response to Reviewer 1 Comments 

Point: Table 1. regarding methicillin and vancomycin resistant isolates, authors replaced the old number with zero; which is not correct. Under VRE heading in Table 1, the percent of vancomycin resistance should be 100%, how it could be 23.6 and later replaced with zero???? 

The same for MRSA. Authors should revise all the cited numbers in this table. 

Response: Regarding the percentage of vancomycin resistance in VRE, your comment is correct, we believe it was a typo mistake. The reported resistance in the referenced article was 100%. The issue with MRSA data was regarding susceptibility to oxacillin, which was reported in the first draft as 4.3%. That average percentage was calculated from three different studies, two of which reported 0% susceptibility and one reported 12.8% susceptibility (Alhussaini MS. . Pakistan J of Biological Sciences. 2016;19(5):233-238, Taha AE, et al. J Infect Dev Ctries. 2022;16(06):1037-1044, Said KB, et al. Diagnostics (Basel). 2023;13(5):819). Similarly, cefoxitin average percentage (11.5%) was calculated from three different studies which reported susceptibility of 21%, 0%, and 13.3%, respectively (Alhumaid S, et al. Ann Clin Microbiol Antimicrob. 2021;20(1):43., Al Musawi S, et al. J Prev Med Hyg. 2022;63(1)., Said KB, et al. Diagnostics (Basel). 2023;13(5):819).  

This variation could be resulted from inaccurate use of the automated susceptibility testing methods. We believe if BMD was used in those studies, we would have gotten more accurate susceptibility results.   

Reviewer 2 Report

the article can be published in this form

Author Response

Thank you for the insightful feedback provided. We hope our study will make a valuable contribution to the field of antibiotic resistance.